# How and under Which Conditions Can We Best Combine Research on School Effectiveness with Research on School Improvement? Establishing Connections Using the Dynamic Approach to School Improvement

Panayiotis Antoniou *, Leonidas Kyriakides and Evi Charalambous

Department of Education, University of Cyprus, 2109 Nicosia, Cyprus
* Correspondence: antoniou.panayiotis@ucy.ac.cy

**Abstract:** This paper aims to discuss how and under which conditions can we best combine research on School Effectiveness with research on School Improvement by utilizing the Dynamic Approach (DA) to School Improvement. We firstly elaborate on the difficulties that exist in trying to merge results from research on School Effectiveness with research on School Improvement. Then, we discuss the need and potential benefit from merging the two research strands, and finally we propose a fruitful merging with reference to the DA framework. Studies that have utilized the DA are also presented and their implications for research, policy and practice are discussed. The last section summarizes the results of the studies that have utilized the DA to highlight conditions that could facilitate a productive merging of the School Effectiveness and School Improvement research.

**Keywords:** school effectiveness; school improvement; merging school effectiveness with school improvement; student outcomes; dynamic approach

## 1. Introduction

This paper aims to discuss the necessary conditions under which we could combine findings from research on School Effectiveness with research on School Improvement. As [1] argues, a significant weakness of the models of educational effectiveness is the fact that they haven't managed to contribute to the improvement of schools and of teaching practice. Research on School Effectiveness and research on School Improvement have been criticized for not creating the necessary synergies towards developing common grounds and approaches that could be applied in their common efforts to improve schools. Such approaches could make use of theoretical frameworks that were empirically found to work with frameworks on teacher development and management of change, explaining how teachers and school stakeholders could best move forward in their attempts to improve schools and promote student learning outcomes [2,3].

Taking this into consideration, in this paper we firstly elaborate on the difficulties that exists in trying to merge results from research on School Effectiveness with research on School Improvement. Then, we discuss the need and potential benefit from merging the two research strands, and finally we propose a fruitful merging with reference to the Dynamic Approach (DA) framework. Studies that have utilized the DA are also presented and their implications for policy, research and practice are discussed.

## 2. Constraints in Merging Research on School Effectiveness with Research on School Improvement

Even though both research strands have common grounds in their efforts to make schools more effective, many have argued that the two traditions have differences in relation to their purposes and aims and their research design. One of the main differences

between the two research strands relates to the type of questions they raise. Particularly, research on School Effectiveness draws its attention on the "What—What best works in improving student outcomes?", focusing on identifying factors that were found to have an impact on student outcomes. Such factors could be related to teacher behaviour/quality of teaching, (e.g., lesson structuring, questioning, student assessment and feedback), with the school policy (e.g., policy on homework, policy on school–parents' collaboration), or even with the national policy (e.g., national curriculum–learning opportunities). On the other hand, research on School Improvement draws its attention on the "How–How can we change our schools so as to improve them?". As [4] argued, the first question focuses on identifying the characteristics of effective schools that were found to have an impact on educational outcomes, whereas the second on the process of change and the necessary procedures towards improving schools.

Further to the research aims and research questions of each strand, there are essential variations, since on the one hand School Effectiveness is focused on the existence of validated theories and explanations (as it is ultimately a research-oriented programme), and on the other hand School Improvement emphasizes problem-solving and how to introduce a change in an educational setting (as it is eventually a programme for innovation) [5]. The different types of questions raised by the two research strands led to differences in the research methodology they utilized. Particularly, most studies under the School Effectiveness strand utilized quantitative research methods and advanced statistical analyses to identify the net effect of several effectiveness factors on student outcomes [6,7]. On the other hand, most studies under the School Improvement strand utilized qualitative research designs aiming to understand the process of change and school stakeholders' priorities, attitudes and perceptions towards the improvement process [8,9].

Despite the differences between the two research strands, a third group of researchers positioned themselves in the middle, trying to merge findings from research on School Effectiveness with some attempts towards School Improvement. As early as in 1993, scholars from School Effectiveness and School Improvement discussed the possibilities of merging these research fields (see for instance the formation of the International Congress for School Effectiveness and Improvement [ICSEI] and Foundation for International Collaboration on School Improvement [FICSI]). In this perspective, [10] with others e.g., [11] have argued about the need to create stronger links between the two traditions. During the last three decades, there have been fruitful attempts to merge School Effectiveness and School Improvement see [12–15].

However, those efforts have sometimes been problematic, and several concerns have been raised in relation to the uncritical adoption of ideas from research and their capacity to be successfully implemented in everyday school settings [16]. Thus, the question that still remains is, can these two areas really be linked, and how and under which conditions? see also [17]. While each one of the two areas has its own theories and relevant findings, there are many common aspects between the two, such as their main purpose of making schools work for all students.

In this paper, we argue that previous attempts and research projects aiming to merge the two traditions had a lack of foundational firmness. This lack of foundational firmness is due to the fact that School Effectiveness research usually studies School Effectiveness via cross sectional studies, by carrying out one or two measurements, whereas School Improvement research usually utilizes longitudinal designs to explore how schools become more effective as time goes by. To resolve this contradiction, we propose the DA towards School Improvement, which attempts to identify the conditions and necessary steps in effectively combining findings from School Effectiveness research with findings from the School Improvement research.

## 3. Establishing a Dynamic Approach (DA) to School Improvement

In this paper, we argue that the dynamic model of educational effectiveness [2] could be a basis upon which a '*a theory-driven and evidence-based approach*' to School Improvement

can be established. A major criticism of educational effectiveness research (EER) is that there are only a few rational models from which researchers can build theory. Nevertheless, during the 1990s, researchers in the area of EER attempted to incorporate the findings of School Effectiveness research and teacher effectiveness research. The resulting integrated theoretical models of educational effectiveness e.g., [2,18,19] have a multilevel structure, but a review of these models revealed that the integrated models did not take into account the dynamic nature of education, and therefore were not able to contribute to the establishment of links between EER and School Improvement [20]. As a consequence, the dynamic model was developed by considering the strengths and limitations of the integrated models and its main aim was to establish links between EER and School Improvement. To achieve this purpose, a multidimensional approach to measure the functioning of the effectiveness factors was introduced see [2]. By using this framework in evaluating the functioning of school factors, one can generate a holistic picture of the functioning of each factor and identify specific improvement priorities. In addition, the situational character of School Improvement factors is taken into account and the importance of taking actions to improve the school learning environment by considering the needs of each school is stressed. A relevant study provided support to this assumption, especially since a reciprocal relation between policy and stakeholders' actions was identified [21,22] provide a historical review of School Improvement research and demonstrate the relations of DA with both the early phase research on School Improvement and the participatory approach. It is beyond the scope of this paper to provide this review, but it is stressed below that the strengths and limitations of these two main School Improvement approaches were taken into account in using the dynamic model to establish the DA.

Researchers, practitioners and policy-makers have widely used the term "theory-driven and evidence-based approach" to emphasize the need for a systematic evaluation of any School Improvement programme by making use of appropriate designs and techniques that can determine its impact on promoting quality and equity in education [23,24]. During the last years, 20 studies as well as two meta-analyses have been carried out to test the validity of the dynamic model. Therefore, these studies have provided empirical evidence supporting the main assumptions of the dynamic model regarding the factors included in the model, as well as the use of the five dimensions to measure the functioning of these factors (for a detailed review of these studies see [25]). The major characteristics of the dynamic model are presented below.

Firstly, the dynamic model is based on the notion that teaching and learning are vigorous procedures that should be continuously adapting to the different needs of teachers and students. Thus, effective education is considered as a dynamic and ongoing process. Second, all studies mentioned above have provided empirical support for the multilevel nature of the dynamic model since factors operating at four different levels (student, classroom, school and system) were found to have an impact on student learning outcomes (see Figure 1). More specifically, the emphasis of the model is on the two main actors of the teaching and learning situation (i.e., teachers and students). The model also refers to specific teacher- and school-level factors which were found to be related with not only cognitive, but also with affective, psychomotor and meta-cognitive learning outcomes [22]. School-level factors were found to have an effect on the teaching-learning situation through the development and evaluation of the school policy on teaching and the policy on creating a learning environment at the school [26]. Third, the model argues that factors operating at the same level (e.g., classroom) are related to each other. Consequently, the concept of grouping of factors is proposed and studies investigating relationships among teacher factors were able to identify stages of effective teaching. Fourth, the model assumes that the school and the context level have both direct and indirect effects on student achievement. Fifth, it is noted that each factor of the model is treated as a multidimensional construct consisting of five dimensions (i.e., frequency, stage, focus, quality and differentiation). By following this approach, not only quantitative but also qualitative characteristics of the factors are considered. Lastly, a distinctive feature of the dynamic model is that it does not

only refer to factors that are important for explaining variation in educational effectiveness, but it also attempts to explain why these factors are important by integrating different theoretical orientations to effectiveness. In this way, a more systematic and comprehensive feedback can be provided to teachers and schools which helps them to identify their own improvement priorities.

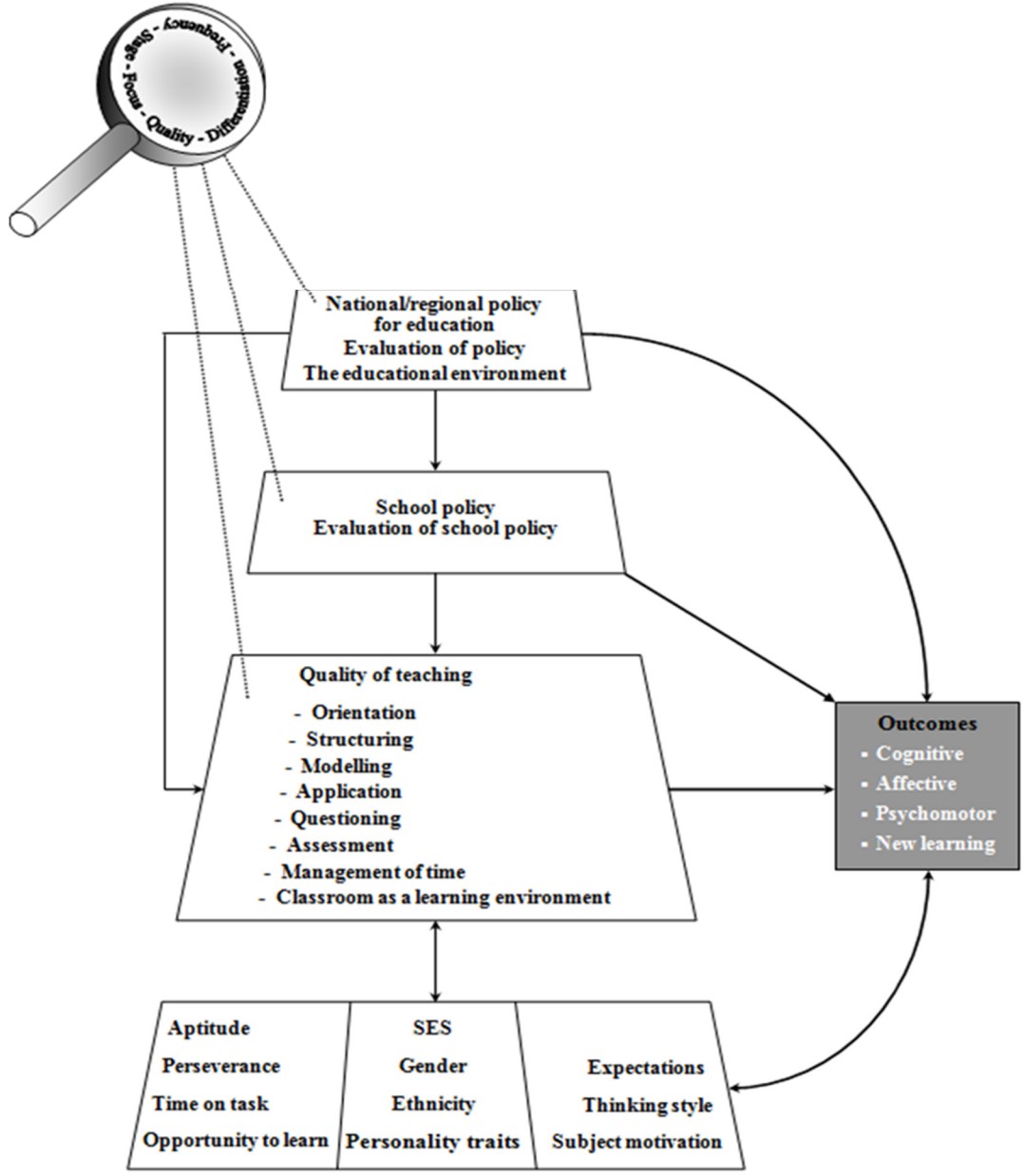

**Figure 1.** The dynamic model of educational effectiveness (adopted from [2]).

Thus, a DA to School Improvement has been developed in order to help schools make use of the knowledgebase of EER included in the dynamic model for improvement purposes. More precisely, the DA was developed by [3] in an attempt to combine important elements of the School Effectiveness research with elements of the School Improvement research in facilitating improvement of student learning outcomes. Together with other researchers in the field of EER (from Cyprus and other European countries), support to the validity of the approach and to its effectiveness was provided as mentioned in the fourth section of this paper. It is also noted that the DA has been implemented in public schools in Cyprus on a voluntary basis, during the last five years. Moreover, one of these

studies was an initiative of the Ministry of Education and was concerned with the use of DA in secondary education [27]. This three-year study provided empirical support about the impact of the DA and based on its results relevant guidelines have been provided by the Ministry of Education to all schools. During the last three years, schools have been encouraged by the Ministry of Education to use the DA in their attempt to establish their own School Improvement strategies and actions. In addition, several European projects have been conducted and policy implications for the participating countries were drawn [28]. It should, however, be acknowledged that more can be done in this domain and in the last section of this paper relevant suggestions for further research are drawn.

Below the main assumptions and features of the DA are presented and recommendations for examining the impact of making use of the DA for improvement purposes are provided.

### 3.1. Assumptions and Main Features of the DA

When policy makers introduce reform policies, various changes are proposed that affect the role of the respective stakeholders, since new ideas and practices are foreseen. However, evaluation studies have identified that over the years these reforms are not successful and are not able to improve the quality of education because they were not initially based on a valid theoretical framework [12]. Thus, the DA promotes the design of School Improvement projects that are based on a theory that has been systematically tested. Specifically, the DA adopts the dynamic model, which accepts the complex nature of educational effectiveness and simultaneously represents the effectiveness factors and their dimensions in a way that improvement of education can be flexibly addressed [29,30]. The dynamic model refers to factors that are important for explaining variation in student achievement and also explains the reasons for which these factors are significant by merging various theories of educational effectiveness. Through this feature, school stakeholders could become aware of the significance of the effectiveness factors involved in their School Improvement initiatives and at the same time of the way these factors operate within a conceptual framework.

Second, the DA supports that student learning should be considered as the most important purpose of any School Improvement initiative. This is due to the fact that learning is the main function of the school. Consequently, schools are expected to take actions for improving their policy for teaching and their policy for creating a learning environment at the school, since these factors were found to be related with student achievement gains. Therefore, schools should be in a position to establish clarity about the general purpose of the improvement project, which should be concerned with the promotion of student learning. Since the DA is based on the dynamic model, which refers to factors that are changeable and associated with student learning outcomes, also the School Improvement effort intermediate aims could address the needs of schools as those have been identified in relation to the functioning of factors of the dynamic model.

Third, the DA is based on the assumption that not all schools are equally effective and therefore assumes that similar strategies for improvement should not be used from all schools in exactly the same way. It is also noted that even for the most effective schools, there is always room for improvement, and consequently these schools should be in a position to identify their improvement areas and design their action plans. The DA considers the dynamic nature of effectiveness and provides support to the utilization of various strategies for improvement from schools, which will be in line with the stakeholders needs, context, knowledge and aims. At the same time, evaluation data should be collected to identify the needs and priorities for improvement of each school.

Fourth, schools are encouraged to develop their own strategies/action plans for improvement. During that process, support and guidance to schools should be offered by an *Advisory and Research Team (A&RTeam)*. This team is important in providing technical expertise and the available knowledgebase on the factors addressed by the school. Schools are considered as professional communities which are indeed responsible for implementing

their own improvement strategies and action plans [31,32]. However, at the same time, schools should not be left alone in implementing the action plans they developed and should be encouraged to make use of the A&RTeam and any other available resource within and/or outside of the school.

Fifth, the DA draws from both the action research paradigm and the experimental research strategy to develop its own improvement strategy. Specifically, the DA does not follow either a strict experimental design approach in introducing and evaluating a School Improvement intervention e.g., [33–35] or the action research project approach [36,37]. Those that follow the strict experimental design approach are usually members of a research team and see themselves as the owners of a School Improvement project. They are also treated as the experts who ask school stakeholders to follow their interventions. They are also responsible for carrying out evaluations for summative purposes and present the intervention results to the research community. This approach has been used in the past but was not found to be successful, mostly because these interventions do not consider the school context and the improvement needs of each individual school [23,38,39]. The DA supports that both the A&RTeam and the school stakeholders should be involved in the design, implementation and evaluation of the improvement initiative so as to avoid some of the difficulties in carrying out experimental studies in which school stakeholders are expected to follow and implement in a rather linear way an intervention which was proposed by others.

In the case of the DA, school stakeholders are those who take decisions on which improvement actions and tasks should be carried out. By using this approach, not only ownership of the improvement project is established but also the stakeholders' experiences and the special situation of each school are taken into account. On the other hand, the A&RTeam has an important role to play in designing the School Improvement strategy. The A&RTeam is expected to share its expertise and knowledge with practitioners and help them develop their own strategies and action plans that are in line with the relevant knowledgebase. This is the main difference between the DA and other School Improvement approaches [40] under the action research paradigm approach, supporting that school stakeholders should be left on their own to develop their own strategies and action plans for improvement [41].

Finally, the DA puts a strong emphasis on the importance of carrying out school evaluation, with a special interest in its formative function, as an important constituent of the School Improvement initiative. As [32] argue, schools should develop their own evaluation mechanisms and make use of such data to improve quality of teaching and the school learning environment. This means that School Self Evaluation (SSE) is a significant aspect of the DA since it is concerned with the assessment of the functioning of the school-level factors of the dynamic model. Thus, the results of the SSE process are expected to provide feedback to the school stakeholders in order to set their priorities for improvement and deal with some of these factors to improve their effectiveness status.

### 3.2. Major Steps of the DA for Effective School Improvement

In this section, the major steps of the DA are briefly presented. Specifically, there are six steps in the DA (see Figure 2), and it is emphasized that school stakeholders and the A&RTeam are expected to be actively involved in each step. This is because their ability to work together and exchange their knowledge and experiences is critical for the success of any School Improvement programme. Readers can find more details for each step of the DA in [3].

The first step of any School Improvement effort is a discussion towards clarifying and fully understanding the purpose and basic aims of the improvement initiative and how these could be achieved. We do acknowledge that it may be difficult to reach consensus between all school stakeholders, albeit this may be crucial for its success. Nevertheless, emphasis should be given in establishing procedures to help school stakeholders under-

stand the aims of School Improvement and more precisely that student learning should be considered as the ultimate aim of their improvement initiatives.

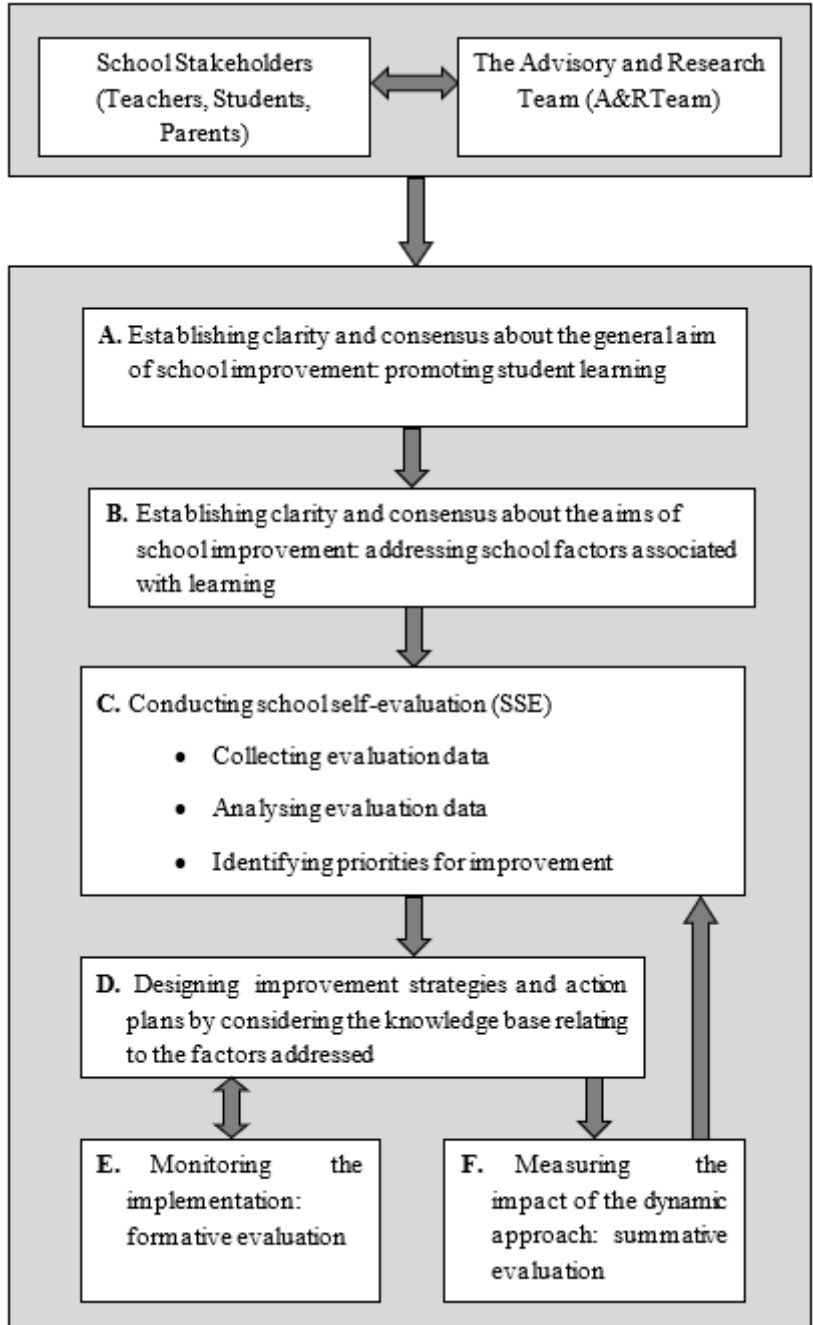

**Figure 2.** The major steps of the dynamic approach (DA) to school improvement (adopted from [6]).

In the second step, the dynamic model and its factors are discussed with school stakeholders. When making use of the DA for School Improvement purposes, school stakeholders are expected to develop strategies and action plans aiming to improve the functioning of effectiveness factors that are situated at the school level of the dynamic model (i.e., the theoretical framework of the DA). These factors, as also mentioned above, are expected to have both direct and indirect effects on student learning outcomes. Designing improvement efforts focusing on the classroom level factors may improve the teaching practice of individuals yet may not necessarily improve the school learning environment. In such cases, teachers who will manage to improve some aspects of their teaching practice

addressed by a specific improvement effort will require, at some stage, another type of support to improve other teaching skills. The DA is based on the assumption that school stakeholders should develop interventions which will not only improve the functioning of the school factors, but ultimately will promote quality of teaching and raise student achievement. Therefore, the DA emphasizes the improvement of the teaching practice but attempts to do so by improving the functioning of the school policy on teaching and of the policy for creating a learning environment. In this way, it is not only learning opportunities that are offered to teachers, but also conditions that enable continuous improvement of teaching practice.

The next step refers to the collection of the evaluation data. The research team and the school stakeholders make use of validated instruments in order to collect data about the functioning of each school-level factors (for a review of these instruments see [3]). The research team then proceeds in analyzing the data and helps school stakeholders identify their priorities for improvement. The improvement areas are then announced to the whole school community and suggestions are given in order to define the specific area/areas of improvement. This step highlights the importance of using an evidence-based approach for School Improvement.

The fourth step is one of the most critical steps of the DA. School stakeholders are expected to consider the available knowledgebase of School Effectiveness research and implement the guidelines emerging from the literature on their school context (with the support of the A&RTeam). At this stage it should be clarified that dynamic model refers to school factors which were found to be associated with student achievement and the conditions under which these factors have stronger effects [5]. In this perspective, the dynamic model refers to qualitative characteristics of the functioning of factors which increase their impact on learning. Schools should therefore draw lessons from the literature on the factors that are addressed and develop their strategies and action plans accordingly. This will be achieved with the support of the A&RTeam who will provide additional input to existing experiences so as to assist schools in generating and implementing their own strategies and action plans.

Then, the fifth step refers to the monitoring of the improvement project implementation via the establishment of formative evaluation mechanisms. Through the establishment of a formative evaluation mechanism, data on the implementation of the action plans could be gathered that could assist school stakeholders in making decisions on how to improve their action plans. The role of the A&RTeam is important, as their expertise in carrying out and analyzing evaluation data could be a valuable resource for schools. At the same time, school stakeholders should be actively involved in the process of formative evaluation. An internal school evaluation system should be developed, through which school stakeholders are encouraged to reflect on their abilities and to improve the functioning of school factors. The results from this formative evaluation could assist school stakeholders in taking informed decisions on improving their action plans [4,13,42]. The establishment of formative evaluation mechanisms is important, since using the available knowledgebase in developing improvement strategies may not necessarily lead to identifying the optimum solution for improving the functioning of school factors. By developing a formative evaluation mechanism, schools could identify the limitations and the weaknesses of their action plans so as to modify them in due time, during the implementation period of the School Improvement intervention, and before it is too late to do so.

The last step refers to assessing the impact of the DA. At this final step of the DA, the A&RTeam and the school stakeholders implement a summative evaluation mechanism. This mechanism aims to evaluate the net impact of the intervention on improving the functioning of the school level factors and on improving student outcomes. This step also stresses the significance of the probability of identifying a new priority area for improvement. More precisely, summative evaluation results may indicate that a school has managed to substantially improve the functioning of the factor(s) addressed, and in this perspective school stakeholders and the A&RTeam could take the decision to identify a new

area for improvement, (e.g., during the next school year) by again collecting evaluation data. Thus, this final step of the DA shows that schools should aim to improve their effectiveness regardless of how efficient they are in each factor, an aspect of the DA that reveals that the dynamic nature of educational effectiveness should be considered in every School Improvement project.

The following section presents the results of studies investigating the assumptions of the DA on promoting quality and equity in education.

## 4. Studies Investigating the Impact of Using the Dynamic Approach for Improvement Purposes

During the last 15 years, a number of experimental studies have been conducted in order to identify the impact of the DA on promoting student learning outcomes and on using the DA to establish School Improvement programmes. Five studies (improvement projects) are discussed below and then the main assumptions of the DA addressed by each study are presented.

The first study investigated the impact of three different approaches to establish SSE mechanisms and investigated their impact on student learning outcomes in mathematics in primary schools in Cyprus [5]. Specifically, two other approaches of School Improvement (beyond the DA) were used to establish school self-evaluation mechanisms for improvement purposes. The first approach was related to the assumption that the school stakeholders are able to generate their own effectiveness criteria and based on that to establish their own school self-evaluation mechanisms. By making use of the evaluation results, school stakeholders can then define their own improvement strategies (participatory approach). The second approach is based on the view of schools as mini political systems with diverse constituencies [43,44]. As a consequence, the concerns of the various stakeholders about establishing school self-evaluation mechanisms to develop School Improvement strategies and actions plans should be initially faced and reduced before encouraging school stakeholders to establish their own school self-evaluation mechanisms for improvement purposes. The added value of using the DA rather than any of the two other approaches to improving student learning outcomes in mathematics was demonstrated, since the impact of DA on student achievement was found to be higher than that of the other two approaches to SSE.

The second study was a quasi-experimental study aiming to examine the impact of the DA on School Effectiveness in secondary education in Cyprus [27]. The added value of using the DA for a longer period and the sustainability of its results one year after its completion was also investigated. More specifically, the DA was employed to six secondary schools in Cyprus, while another group of schools with similar characteristics comprised the control group. The data analysis revealed that schools in the experimental group achieved better results compared to those in the control group at the end of each school year of the project. Students' achievement gains in mathematics were even bigger for schools that made use of DA for the three-year period. In relation to the sustainability of the DA, it was found that schools which made use of it for only one year managed to achieve better results than the control group, not only at the end of the first year, but also two years after the implementation of the intervention (i.e., at the end of the project).

The third study was carried out in five European countries to investigate the impact of the DA on confronting and reducing bullying in primary schools [28]. It was found that schools which made use of the DA were able to reduce bullying at a significantly higher level than those in the control group. This finding emerged from both cross- and within-country analyses.

The last two projects presented in this section aim to investigate the impact of the DA on promoting both quality and equity in socially disadvantaged schools. More precisely, the fourth study [45], which took place in primary schools in Cyprus, was the starting point for the attempt to investigate the impact of the DA on promoting quality and equity in a specific school setting. The results revealed that the experimental schools managed to

promote student achievement in mathematics to a greater extent than schools in the control group. It was also found that the direct effect of SES on achievement in mathematics was reduced in the experimental schools.

Finally, the fifth study aimed to support primary schools in socially disadvantaged areas in four European countries by using the DA to promote students' basic skills in mathematics [46]. In each country, the DA was found to have an effect on promoting student learning outcomes [47]. Implications can be drawn with regard to measuring the effectiveness status of schools in terms of the equity dimension. We argue here for the value of collecting longitudinal data in order to establish formative school evaluation mechanisms and measure changes in the impact of background factors on student achievement over time [38]. In this study, the sustainability and the value added of offering the DA to schools for more than one year in improving student achievement in mathematics (quality) and reducing the impact of SES on achievement (equity) was also investigated. Schools that made use of DA for only one year were found to outperform schools of the control group not only at the end of the first year of the intervention but also at the end of the second year. This finding provides support to the sustainability effect of DA to those who made use of DA for only one year.

The above studies provide support to the main assumptions of the DA. The first main assumption concerns the impact of the DA on promoting student learning outcomes (cognitive and affective) and this has been tested through all the aforementioned studies. It is stressed here that studies 1, 2, 4 and 5 have shown that the DA can promote student achievement gains in mathematics, whereas study 3 has revealed the impact of the DA in helping schools reduce the number of bullying incidents. These results provide some support to the argument that the DA can be used to promote different types of student learning outcomes. Further studies investigating the impact of the DA on promoting student achievement gains in other school subjects and in other types of learning outcomes are, however, needed. In addition, these studies/interventions took place mainly in primary schools. However, one of the studies presented above took place in secondary education and some empirical support that DA can be used to promote student learning outcomes in compulsory education has been provided. Further studies testing the generic nature of DA are however needed. These studies may also reveal differential effects of DA on promoting different types of learning outcomes in different phases of education. The second assumption tested was the sustainability of the DA, and study 2 as well as study 5 have demonstrated that one year after the implementation of the DA, the approach was found to have had an impact on student learning outcomes in mathematics. These two studies have also managed to examine the third assumption of the DA, which is the added value of offering the DA for more than one school year. This was demonstrated since the impact of the DA on student achievement in mathematics (study 2 and study 5) as well as in reducing the impact of SES on student learning outcomes (study 5) was greater in schools which made use of the DA for more than one school year. It is emphasized here that these two studies are the only ones that have managed to test the second and third assumptions of the DA, since in all the other studies the DA was employed for only one school year. The fourth assumption of the DA concerns its indirect effects on cognitive and affective outcomes. Since the DA has to do with the improvement of the school level factors of the dynamic model, it is expected that through this whole school intervention, not only will the functioning of these effectiveness factors improve, but student learning outcomes will also be enhanced as a result of the impact that the intervention can have on improving the functioning of the school factors. This assumption was tested through four studies (see studies 2, 3, 4 and 5). The DA was therefore found to have an effect on student achievement gains in mathematics [25] and on reducing bullying [22] through improvement in the functioning of school factors.

It is also important to note that two of these studies (i.e., studies 3 and 5) took place in different European countries, and in each of these countries indirect effects of the DA were reported. This reveals that the DA can be applied in different educational systems (at

least within Europe) and in different educational contexts (including schools situated in socially disadvantaged areas) in order to promote quality in education by improving the functioning of school factors. Lastly, as regards to the fifth assumption, which is about the impact of the DA on the equity dimension of School Effectiveness, the last two studies have shown that the direct effect of SES on student achievement in mathematics was reduced only in the schools that made use of the DA (studies 4 and 5). In addition, the total effect of SES on final achievement in mathematics was observed to increase in schools in the control group, whereas it remained the same in schools which made use of the DA (see study 5).

Taking all the above into consideration, the last section of this paper summarizes the results of the studies described above to highlight the necessary conditions that could facilitate a productive merging of the School Effectiveness and School Improvement research.

## 5. Establishing Connections between Research on School Effectiveness with Research on School Improvement: Implications for Research, Policy and Practice

In this paper we propose a broadening of the agenda of research on educational effectiveness and improvement and claim that research on developing and testing theories of educational effectiveness can contribute significantly to establishing an effective approach to School Improvement. In addition, this paper supports the claim for an evidence-based and theory-driven approach to School Improvement. The importance of using the dynamic model to establish such an approach to School Improvement is stressed. More precisely, the dynamic model of educational effectiveness has been used as the theoretical framework (i.e., providing the factors/components that were found to have an impact on student learning outcomes) in developing the DA. In addition, the knowledgebase of EER is taken into account in establishing the handbook that is expected to help schools design their own School Improvement plans. What is important to mention is that the DA utilized at the same time the theoretical framework of the dynamic model, from the School Effectiveness strand, as well as important elements/processes of the School Improvement strand (such as the active involvement and critical reflection of the participants in identifying their own priorities and needs for improvement, the provision of support and feedback, the development of various action plans according to various needs, regular meetings for reflection and discussion, and formative evaluation processes). Therefore, the main findings of School Improvement research are taken into account, and especially the construct of ownership is considered in establishing the DA. In this perspective, the main assumptions, features and steps of this DA are discussed. We also present the main results of five projects measuring the impact of the DA on promoting student achievement. From the research projects that have utilized the DA, we discuss the conditions under which the DA could be effectively used in a way that can bridge the gap between research on School Effectiveness and research on School Improvement. Suggestions for further research are also outlined below.

First, the DA is not a top-down approach since it can only be used by those teachers and schools who share the value assumptions of the dynamic model. Specifically, teachers and schools participating in improvement projects based on the DA should treat the promotion of quality and equity in education as the ultimate aim of their interventions. This implies that teachers and schools participating in improvement projects expect that teachers and schools can have a significant effect on promoting student learning outcomes. By participating in school interventions, their knowledge, skills and practices could be enhanced. This also implies that teachers and schools need to acknowledge that there is room for improvement, and that they cannot remain effective without taking any actions towards their improvement [48]. Moreover, school stakeholders participating in projects based on the DA should recognize the importance of effectiveness factors included in the dynamic model to take actions for improving the functioning of these factors. These two preconditions are addressed in the first two steps of the DA. The main aim of these two steps is to develop consensus about the ultimate purpose of a School Improvement intervention (i.e., promotion of quality and equity) and its intermediate aims (i.e., improvement of specific teacher and/or school factors). It should be acknowledged here that studies using

the DA revealed that not all schools and teachers can be convinced of the need to share these value assumptions [49] and it is for this reason that teachers and schools should be given the right to decide whether they are prepared and willing to participate in an improvement project based on the DA. Therefore, this improvement approach cannot be imposed by all policymakers or school administrators. DA could only have benefit on those who share its value assumptions and objectives (see steps A and B of the DA).

Second, if we are to effectively merge research on School Effectiveness and School Improvement, a consensus should be agreed on the importance of the school initial, on-going and final evaluation. More precisely, the DA reveals the importance of an initial measurement of the functioning of school factors, which is expected to help school stake-holders identify and focus on their immediate and most important needs and priorities for improvement ([50]. The experimental studies which made use of DA for improvement purposes revealed that this initial measurement should only be conducted for formative reasons and the A&RTeam should make sure that nobody can use this data for summative purposes [51]. Specifically, all teachers at a participating school should be invited to complete a questionnaire measuring the functioning of each school-level factor [3]. The A&RTeam will then analyze the data and rank the functioning of the school factors for each individual school. In this way, it is possible to identify those factors that perform less well in each school. Then, the A&R Team will present the results of this analysis to a staff meeting and each school will identify the combination of factors (among those that perform less well) that they would like to address. In this way, each school can define its own improvement priorities. As a result, a more focused intervention is expected to take place in each school, which does not address all school factors but only those that are underperforming and have been agreed by all school stakeholders. Therefore, the DA is based on the assumption that the use of a whole School Improvement intervention has to be concerned with the actual needs of each school in order to have an impact on improving the functioning of school factors and through that to promote student learning outcomes.

Third, in implementing the third and the fourth steps of the DA, schools should receive external support from the A&RTeam. This team is expected to share its expertise with teachers and other school stakeholders, for example when measuring the functioning of factors and analyzing the data in order to help in identifying a school's priority areas. In addition, the A&RTeam's role is to support schools in developing their own action plans and implementing them. To achieve this aim, a handbook is provided with suggestions for activities that might be undertaken to improve the functioning of each factor. It is, however, up to the teachers and schools to define the tasks that they would like to implement and the period during which these tasks would be carried out. Therefore, teachers are offered support in the form of the handbook, but at the same time they are free to decide whether the suggested activities can be adapted to their own situation and incorporated into their own action plans.

It is stressed here that all studies investigating the impact of DA revealed that during a School Improvement intervention, the impact of the DA on student learning outcomes depends on the effort that individual teachers and schools put into implementing their action plans [5]. For this reason, regular meetings should be organized with teachers and schools (approximately one every six weeks) to help them reflect on the implementation of their action plans and identify ways to further develop these plans. In this way, teachers and schools are not only provided with systematic support to help them implement their action plans but are also kept on track with respect to implementing their intervention throughout the school year [51]. Teachers and schools do not only receive support when implementing their improvement strategies and action plans at the beginning of the inter-vention, but the A&RTeam becomes a partner of each individual school throughout the implementation period (e.g., a school year). The A&RTeam may also need to respond to specific challenges that schools may face during the implementation of their action plans. For this reason, the A&RTeam should encourage schools to get in touch with them when a specific challenge/problem arises which might need immediate action rather than expect-

ing this problem to be reported in the next session. One can therefore argue that the DA assumes that schools receive three different types of support: (a) a handbook presenting the framework and rationale of the DA, including specific actions that can be taken to improve the functioning of the factors; (b) the establishment of a network for these schools that are addressing similar areas requiring improvement which will help them to learn from each other; and (c) the organization of regular meetings with the A&RTeam to help schools better understand how the factors addressed can have an impact on learning. During the implementation of improvement projects based on the DA, it has also been found that teachers and schools may need specific resources to implement their action plans and the A&RTeam should find ways (in cooperation with other stakeholders) to offer these resources to schools. This implies that the DA raises awareness of factors promoting learning and encourages school stakeholders and the A&RTeam to identify ways to improve the classroom, the school and the home learning environment not only by developing and implementing appropriate policies, but also by generating prerequisites for learning.

Although the success of an intervention is mainly based on the effort that teachers and schools are prepared to put into implementing their action plans, it is also clear that the A&RTeam has an important role to play in helping schools and teachers to define their priority areas and devise their action plans, and also in supporting them when implementing their plans. One of the essential differences between the DA and other approaches to School Improvement has to do with its attempt to consider effectiveness as a dynamic rather than a stable construct. Therefore, there is expected to be variation in the effectiveness status (rather than time stability) of teachers, schools and systems. Moreover, DA assumes that changes in the functioning of school factors can explain changes in the effectiveness status of the schools. This implies that schools should identify their weaknesses and develop specific policies in order to improve teaching and, through that, achieve better learning outcomes.

In the last part of this section, we outline the main messages of this paper, which is an attempt to propose a broadening of the agenda of research on educational effectiveness and improvement. This paper claims that research on developing and testing theories of educational effectiveness can contribute significantly to establishing an effective approach to School Improvement. Specifically, it is shown that studies testing the validity of the dynamic model contributed to the establishment of the DA. At the same time, studies investigating the impact of DA contributed to the development of a theory of educational effectiveness that takes into account the dynamic nature of education. From the research projects that have utilized the DA, it was also possible to identify the conditions under which the DA can be used to promote quality and equity in education. Three main lessons emerged from these projects. First, the studies revealed the limitations of top-down improvement approaches and demonstrated that DA can only be used by those teachers and schools who share the value assumptions of the dynamic model. Therefore, this approach cannot be imposed by policymakers or school administrators. This argument is in line with the first two steps of the DA and revealed the importance of establishing consensus among school stakeholders about the general aim and the main objectives of a School Improvement strategy. Second, we argue for the importance of school initial, ongoing and final evaluation. Findings of initial evaluation can help school stakeholders identify those factors that need to be addressed in their own school. As a result, a more focused intervention is expected to take place in each school, which does not address all school factors but only those that are underperforming and have been agreed by all school stakeholders. Ongoing evaluation is based on the assumption that action plans should be continuously developed by considering the difficulties and challenges that school stakeholders are likely to face during the implementation phase. Final evaluation at the end of the school year is not only conducted for measuring the impact of an intervention on the functioning of school factors and through that on student learning outcomes; its main contribution is to help schools identify new improvement priorities and further develop their own School Improvement strategies and action plans for the following year.

This argument is in line with the assumption of DA that schools should be continuously involved in School Improvement interventions, and empirical support has been provided by studies demonstrating the added value of using DA for more than one school year. Third, this paper draws attention to the critical role of both school stakeholders and the A&RTeam in implementing DA. As it has been argued above, the success of an intervention is based on the effort that teachers and schools are prepared to put into implementing their action plans, but at the same time the A&RTeam has an important role to play in helping schools and teachers to define their priority areas and devise their action plans, and also in supporting them when implementing their plans.

In this paper we also discussed some of the main difficulties and the importance of combining research on School Effectiveness with research on School Improvement. We have also stressed the importance of the DA and described some of the basic conditions needed to establish connections between the two research strands. Even though we have presented research findings supporting the effectiveness of this approach in student outcomes, more studies should be conducted in different countries to test the generalizability of the results and to illustrate how the DA can be used by policy and practice towards School Improvement. More qualitative studies (e.g., school case studies) could also be conducted to explore and clarify the difficulties and barriers that schools face in implementing the DA (see for example [49]). Using this framework might also help us consider the dynamic character of effectiveness, since not only schools that have managed to improve, but also schools that have not manage to make any progress or have even declined in their performance, could be identified. Such studies could encourage the research community to reconsider the relation between research on School Effectiveness and School Improvement and in this way, stronger links between research, policy and improvement of teaching practice can be established.

**Author Contributions:** Conceptualization, P.A., L.K. and E.C.; methodology, P.A., L.K. and E.C.; formal analysis, P.A., L.K. and E.C.; investigation, P.A., L.K. and E.C.; resources, P.A., L.K. and E.C.; data curation, P.A., L.K. and E.C.; writing—original draft preparation, P.A., L.K. and E.C.; writing—review and editing, P.A., L.K. and E.C.; project administration, P.A., L.K. and E.C. All authors have read and agreed to the published version of the manuscript.

**Funding:** This research received no external funding.

**Institutional Review Board Statement:** Not applicable.

**Informed Consent Statement:** Not applicable.

**Conflicts of Interest:** The authors declare no conflict of interest.

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
