# Peer review of "How and under Which Conditions Can We Best Combine Research on School Effectiveness with Research on School Improvement? Establishing Connections Using the Dynamic Approach to School Improvement"

_education, doi:10.3390/educsci12080537_

Round 1
Reviewer 1 Report
I greatly appreciate the work of Kyriakides and his research team. Great merit with the DMEE and also with the
DASI.
However, I come to a different assessment or conclusion in some sections. This is to be understood as a critique of the DASI in academic discourse. For this reason, I do not think it makes sense to make demands in the classic style of a review as to what should/must be revised. This position of power does not suit me in a case where both positions are established and have their justification.
I see my critical questions and comments primarily as an offer to the authors to examine, reflect and question their own arguments in order to be able to further develop their own model.
Accordingly, this review should be understood more as a recommendation.
However, I would be happy to exchange views if this is desired.
Overall, I think it is important to clarify the link between school effectiveness and school improvement. Is the DASI about schools using the DMEE in practice? For me, that would be a link between SER and SI practices, i.e. a practical perspective. Or is it about combining the two research approaches and thus the central questions raised in the text at the beginning, in order to gain new insights into the link between the two?
My impression is that DASI wants to combine the first, SER and SI practices. Otherwise, it would also need a theoretical model for changing school processes and that would also have to be the subject of research.
However, this should also be clearly stated in the article.
In addition, as mentioned in the article, the DMEE and Da focus on student outcomes. This is also an important factor. In an ever more rapidly changing world, schools are called upon to adapt to external challenges more than ever before. If school improvement is understood as the ability of schools to adapt to such changes (Maag Merki et al. 2021, Feldhoff, et al. 2021). These challenges can be related to student outcomes. However, there are many challenges that are not directly relevant to student outcomes but to schools and students.
Adaptivity is not addressed by the DMEE and DA. Nor can it, because the models are not based on a theoretical concept of change in school processes. This is not necessarily to be understood as a criticism but marks the limits of the model. It would also be helpful for the linking debate to say which aspects the models can achieve and which they cannot. Here, in the end, a corresponding classification is missing once again
Further comments and questions can be found in the manuscript.

Reviewer 2 Report
I believe that due to the high plagiarism ratio of this manuscript the review process cannot further proceed. In more detail, the software iThenticate revelead a plagiarism ratio of 36% which is not acceptable due to publication ethics. Furthermore, there are 11 papers with over 60 words of similarity which is not acceptable due to the international publication standards. Thus I reject this paper and I propose the authors rewrite their manuscript and resubmit it free of plagiarism issues!

Reviewer 3 Report
You state on Line 131 - “a DA to school improvement has been developed”. Mention by whom, in which year, to which version you refer, whether it has moved from the research phase to the public policy phase, whether it has been taken up in public and educational policies at the local, national or international level.
Briefly situate the DA model in a historical perspective: which models preceded it, which factors led to its emergence, why there was no systematic concern to combine RSE with RSI?
Contextualise the analysis: level of education (primary, secondary, high school, university), country / group of countries / continent, respectively private / state education.
Give some examples for “Other school improvement approaches”, except DA (line 211)
Specify the impact of social, economic or generational transformations on implementing the DA model
Specify in the text the meaning of the acronyms ICSEI, FICSI, EER
Specify more clearly what concrete elements of novelty this article brings - (see sections 3 and 4, figure 1 inclusive) compared to the article “Using the dynamic approach to school improvement to promote quality and equity in education: a European study”. Is the DA model in fact the DASI model, analysed in that article?
Reviewer 4 Report
The manuscript has got a theoretical design. It is regarded o two basic concepts and authors presented empirical studies in the five subchapters, which are presented in the logical order. Author wrote whole manuscript in the understandable form. Authors adhered all conditions regarding to write this kind of study.
I am thinking only about it, that authors sused many relatively older studies, some are older than twenty years. And these studies created more than half of all used (research) studies. Maybe, authors could add some newer resources. if they are not existing, so it could be summarized in the conclusion of the manuscript and also to write reasons, why this is not reflected nowadays and add some suggestions.
On the basis of previous comment, it is clear, that authors add conclusion, where also some summarization should be mention and some ideas written for the aiming of these concepts.
I hope my comments are helpful.
Round 2
Reviewer 2 Report
Overall, the revised manuscript adds and improves upon the first. I was generally satisfied with the author's response to my questions and comments.